# Microbial Diversity and Flavor Regularity of Soy Milk Fermented Using Kombucha

**DOI:** 10.3390/foods12040884

**Published:** 2023-02-18

**Authors:** Xinhui Peng, Qiang Yue, Qianqi Chi, Yanwei Liu, Tian Tian, Shicheng Dai, Aihua Yu, Shaodong Wang, Huan Wang, Xiaohong Tong, Lianzhou Jiang

**Affiliations:** 1College of Food Science, Northeast Agricultural University, Harbin 150030, China; 2Specialty of Food Science and Engineering, Heilongjiang Open University, Harbin 150000, China; 3Heilongjiang Beidahuang Green Health Food Co., Ltd., Jiamusi 154000, China; 4College of Agriculture, Northeast Agricultural University, Harbin 150030, China; 5College of Food Science and Engineering, Hainan University, Haikou 570228, China

**Keywords:** fermentation, soy milk, gene sequencing, microbial diversity, flavor regularity

## Abstract

Plant-based milk is considered a healthy and environmentally sustainable option. However, due to the low protein content of most plant-based milk and the difficulty of gaining flavor acceptance by consumers, its production scale is usually limited. Soy milk is a kind of food with comprehensive nutrition and high protein content. In addition, kombucha is naturally fermented by acetic acid bacteria (AAB), yeast, lactic acid bacteria (LAB), and other microorganisms, and the microorganisms in its system can improve the flavor characteristics of food. In the present study, LAB (commercially purchased) and kombucha were used as fermenting agents for soybean, which was used as a raw material to produce soy milk. A variety of characterization methods were used to study the relationship between the microbial composition and flavor regularity of soy milk produced with different proportions of fermenting agents and different fermentation times. In soy milk produced at 32 °C with a mass ratio of LAB to kombucha of 1:1 and a fermentation time of 42 h, the concentrations of LAB, yeast, and acetic acid bacteria in the milk were optimal at 7.48, 6.68, and 6.83 log CFU/mL, respectively. In fermented soy milk produced with kombucha and LAB, the dominant bacterial genera were *Lactobacillus* (41.58%) and *Acetobacter* (42.39%), while the dominant fungal genera were *Zygosaccharomyces* (38.89%) and *Saccharomyces* (35.86%). After 42 h, the content of hexanol in the fermentation system of kombucha and LAB decreased from 30.16% to 8.74%, while flavor substances such as 2,5-dimethylbenzaldehyde and linalool were produced. Soy milk fermented with kombucha offers the opportunity to explore the mechanisms associated with flavor formation in multi-strain co-fermentation systems and to develop commercial plant-based fermentation products.

## 1. Introduction

In recent decades, there has been a growing demand for the development of foods with greater health benefits and environmentally friendly manufacturing processes. In this respect, researchers are increasingly interested in plant-based food products, especially various types of plant-based milk, among which soy milk has become a popular commercial product with an expanding market size. Recently, to improve the nutrient bioavailability, oxidation resistance, and flavor attributes of soy milk products, fermentation has gradually become a new processing and preservation approach [1]. At present, the preparation method of fermented soy milk mainly includes selecting soybeans, grinding them into soy milk, adding ingredients, homogenizing, sterilizing, inoculating, fermenting, and ripening after refrigeration. The quality of fermented soy milk is affected by many factors, including the fermentation agent, time, and temperature. For example, Wang et al. [2] found that the bacterial species and fermentation time had significant effects on the growth of bacteria and the physicochemical properties of the resulting soy milk, with the difference in the concentration of different bacterial species reaching 9 × 10^8^ CFU/mL. In addition, hexaldehyde and glutaraldehyde (the main source of the beany flavor of soy milk) were produced via the peroxidation of unsaturated fatty acids catalyzed by lipoxygenase, and fermentation can degrade some aldehydes. For example, Huang et al. [3] reported that co-fermentation (lactic acid bacteria combined with *Kluyveromyces marxianus)* can effectively degrade the aldehydes in goat milk and produce organic acid compounds.

With the development of bioinformatics tools, new techniques for food flavor analysis and microbial characterization are emerging. Indeed, next-generation sequencing (macro genomics, flavor omics, etc.) and flavor analysis (electronic tongue, gas chromatography–mass spectrometry (GS-MS), etc.) technologies have played an important role in exploring the mechanisms associated with the formation of characteristic flavor substances and clarifying the contribution of microorganisms to this process. For example, Zheng et al. [4] determined the microbial composition and flavor characteristics of handmade cheese using gene-sequencing technology, finding that *Kluyveromyces* and *Sporothromyces* were associated with the formation of 2-decenaldehyde, 2,3-butanediol, phenylethanol, and some amino acids, while *Lactobacillus* was associated with the production of ethanol, butyric acid, acetic acid, and alanine. Jiang et al. [5] also applied high-throughput sequencing (HTS) technology and GC-MS spectrometry to analyze the relationship between volatile flavor substances and the makeup of the microbial community in cow’s milk. The results showed that the concentration of *Lactobacillus* in most samples was the highest, ranging from 41.6% to 98.3%. In addition, *Lactobacillus* was also significantly correlated with the levels of benzaldehyde, 2,3-pentanedione, ethanol, and ethyl acetate.

Kombucha is a fermented black tea drink produced from a symbiotic culture of yeast and bacteria. During fermentation, the microorganisms in kombucha produce amino acids, polyphenols, organic acids, and other nutrients [6]. In studying *Saccharomyces cerevisiae* isolated from kombucha, Hirst et al. [7] found that many biochemical reactions occur during the fermentation process, including the formation of alcohols and aldehydes. However, the concentration of LAB in kombucha is extremely low. Marsh et al. [8] employed high-throughput sequencing technology to investigate the composition of the microbial community in kombucha and found that the two most dominant bacterial genera were *Acetobacter* and *Gluconacetobacter*, accounting for up to 90% of the total bacterial population. This leads to fermented kombucha products with strong acidity, a simple taste, and low palatability.

LAB are important functional bacteria in the natural fermentation of food. They not only metabolize the chemical components of raw materials, thus improving the nutritional value [9], but also produce low-threshold volatile compounds such as aldehydes, esters, alcohols, and ketones, increasing the flavor [10]. In order to enhance their functionality, milk, cheese, and coffee fermented with kombucha or LAB have been commercially produced [11]. However, their mixtures have not been used commercially as soy milk starters, so detailed research is needed to evaluate their potential for producing this type of food.

In the present study, LAB were combined with kombucha to ferment soybeans, with HTS and GC-MS used to assess the microbial composition and flavor-producing mechanisms in fermented soy milk produced with different fermentation times and different proportions of the two fermentation agents. In contrast to single-strain fermentation, this study covers a wider range of microorganisms, including yeast, lactic acid bacteria, acetic acid bacteria, and other common strains. In addition, the fermentation of soy milk with kombucha can not only improve the rheological properties of soy milk and increase its contents of vitamins and aglycone isoflavone but also promote the formation of unique flavor and increase consumer satisfaction with its taste and flavor. Therefore, this study can more accurately clarify the relationship between the dominant strains and flavor substances in soy milk. The flavor analysis of plant fermentation products manufactured using symbiotic starter cultures is a developing field of research, and the present study is expected to provide important foundational information for this field.

## 2. Materials and Methods

### 2.1. Materials

Kombucha was purchased from Shaanxi Wanzi Biological Co., Ltd. (Xi’an, China). Fructooligosaccharides (FOS; 99% food-grade) were obtained from Shanghai Mingyu Biotechnology Co., Ltd. (Shanghai, China). Soybeans (CNA20140575.5) were purchased from Heilongjiang Beidahuang Food Co., Ltd. (Heilongjiang, China). Lactic acid bacteria (LAB), including *Lactobacillus acidophilus*, *L. bifidus*, *L. rhamnosus*, and *L. casei* (commercially purchased, 1:1:1:1), were cultured at Xi’an Dongfeng Biotechnology Co., Ltd. (Shaanxi, China).

### 2.2. Preparation of Fermented Soy Milk

The soybeans (50 g) were soaked in water (*w*/*v* = 1:3) for 12 h at 4 °C. The soaked soybeans were then added to 150 mL of 0.3% (*w*/*v*) sodium bicarbonate solution at 80 °C, kept warm for 5 min, and stirred for 3 min with a soybean-to-water ratio of 1:8 (*w*/*v*). The soy milk was then collected. FOS and sucrose were then added to the sample (4.5% *w*/*v*), and the mixture was homogenized (20 MPa, 3 min). The soy milk was sterilized (115 °C, 15 min) and cooled to room temperature. Different secondary activation solutions (4% *v*/*v*) were inoculated into soy milk samples in turn and cultured at 32 °C in an incubator to obtain fermented soy milk with different fermentation times. The inoculation sequence of the strains was *Lactobacillus acidophilus*, *L. bifidus*, *L. rhamnosus*, *L. casei*, and kombucha strains. Fermentation was terminated when pH reached 4.4. In total, six treatment groups were cultured and analyzed: (1) unfermented soy milk (denoted as sample S), (2) soy milk fermented with LAB only (sample L), (3) soy milk fermented with kombucha only (sample K), (4) soy milk fermented with LAB and kombucha at a mass ratio of 2:1 (sample LK2.0), (5) soy milk fermented with LAB and kombucha at a mass ratio of 1:1 (sample LK1.0), and (6) soy milk fermented with LAB and kombucha at a mass ratio of 1:2 (sample LK0.5).

### 2.3. Analysis of the Microbial Community

#### 2.3.1. Dynamic Growth Curves

Total DNA was extracted from 100 μL of the fermented soy milk samples using a tissue kit (Qiagen, Hilden, Germany), and standard curves were established following the protocols for the kit. The standard curve was generated using the ITS rRNA gene of *Pichia pastoris* and the 16S rRNA genes of *Komagataeibacter xylinus* and *Lactobacillus sakei*. A StepOnePlus Real-time PCR System was used to conduct absolute qPCR analysis of the yeasts, AAB, and LAB in the samples following the method reported by Xiao et al. [12]. RNA was reverse-transcribed into CDNA, CDNA was added to the real-time PCR dye PreMIXA2010A0112, and then samples were added to the qPCR Array. The microbial growth curves were calculated using the following modified Gompertz equation.
(1)$$\log N(T)=\log\frac{N_{\max}}{N_{0}}\times \exp\left\{-\exp\left[\frac{\mu_{\max}\times 2.718}{\log(N_{\max}/N_{0})}\times (L-T)+1\right]\right\}$$
where *N_(T)_* is the cell number at *T*; *N*_max_ is the maximum cell number; *N*_0_ is the initial cell number; *μ*_max_ is the specific growth rate; *L* is the lag phase; and *T* is the sampling time.

#### 2.3.2. Genomic DNA Extraction for Bacteria and Fungi

Aliquots (1.5 mL) of the fermented soy milk samples were centrifuged (1500× *g*, 4 °C, 10 min), and 567 μL of TE buffer (BSAST Inc., Beijing, China) was added. Next, 15 μL of 20 mg/mL proteinase K (HNEB Inc., Hubei, China) and 30 μL of 10% SDS (SIGMA, sigma aldrich, SL, USA) were added, followed by incubation at 37 °C for 1 h until bacterial lysis was complete. Following this, 100 μL of 5 mol/L NaCl and 80 μL of CTAB NaCl solution were added, followed by incubation at 65 °C for 10 min. Tris-saturated phenol/chloroform/isoamyl alcohol (LMAI Inc., Shanghai, China, 25:24:1 *v*/*v*/*v*) was used to extract the DNA, and isopropanol was used to precipitate it. The DNA was dissolved in 50 mL of TE buffer after washing and precipitation with 70% ethanol once. RNase A (MERCK, NJ, USA, 1 mL, 10 mg/mL) was added, and then the mixture was stored at 4 °C until analysis [13].

#### 2.3.3. PCR Amplification and High-Throughput Sequencing

Diluted genomic DNA was used as a template, and specific primers with barcodes were used in accordance with the selection of the sequencing region. High-fidelity enzymes (New England Biolabs, Ipswich, MA, USA) were used to conduct the PCR analysis. For the bacteria, the primer for the 16S V4 region was 515F-806R, and that for the ITS1 region was ITS1F-ITS2. For the fungi, the primer for the 6S V4 region was 515F-806R, and that for the ITS1 region was ITS1F-ITS2. Phusion R High Fidelity PCR Master Mix (15 mL) with GC buffer, forward and reverse primers (0.2 mmol/L), and 10 ng of the template DNA were added to 30 mL of the PCR system, with the following reaction process: pre-denaturation at 98 °C for 1 min, 30 cycles of denaturation at 98 °C for 10 s, annealing at 50 °C for 30 s, and extension at 72 °C for 60 s, followed by holding at 72 °C for 5 min [14]. The product was recycled with a gel-cutting recovery kit, while a library-building kit was used to construct the library. The resulting library was quantified using Qubit. After passing the test, MiSeq was used for online sequencing.

#### 2.3.4. Bioinformatics Analysis

The offline data obtained using the Illumina MiSeq/HiSeq sequencing platform were pre-processed to remove low-quality data, including the primer sequences, unmatched sequences, and sequences that were too short. The species were classified using UPARSE software V7.1 (Illumina, San Diego, CA, USA) after obtaining representative sequences for each operational taxonomic unit (OTU) [15].

### 2.4. Determination of Flavor

#### 2.4.1. Electronic Nose and Tongue Analysis

The fermented soy milk samples were diluted two-fold, and 10 mL was placed in 120 mL sample bottles for odor detection using an electronic nose with 10 metal oxidation sensors. Data were collected at 1 s intervals for a total of 60 s. An electronic tongue was also used to identify eight taste characteristics (sourness, bitterness, astringency, aftertaste-B, aftertaste-A, umami, richness, and saltiness). The electronic tongue was calibrated before use. Each signal acquisition time was 120 s, and the cleaning sequence was set before testing each sample [16]. Alpha Soft V14.2 (ALPHA MOS, Toulouse, France) was used for data processing.

#### 2.4.2. Volatile Flavor Component Analysis

Before analysis, solid-phase microextraction (SPME) fiber was treated at 250 °C for 1 h until the baseline was stable. Aliquots (10 mL) of the fermented soy milk samples (fermented for 42 h) were put into a headspace vial and balanced at 45 °C for 25 min. The SPME fiber was then added and held at 45 °C for 40 min to absorb the flavor volatiles. The SPME fiber containing the headspace analytes was then inserted into the GC injector port (250 °C) for 1 min for desorption. The desorbed flavor compounds were separated in a single-quadrupole GC-MS instrument (GCMS-QP2020; Shimadzu Corp., Kyoto, Japan) equipped with an HP-5 capillary column (30 m × 0.25 mm, 0.25 μm). The GC oven temperature started at 35 °C, where it was held for 3 min, before increasing at 5 °C/min to 200 °C and then at 10 °C/min to 230 °C, where it was held for 10 min. The carrier gas was helium (He) at a flow rate of 0.80 mL/min, with no shunt injection. The ion source temperature was 200 °C, the emission current was 200 μA, and the detection voltage was 350 kV [17].

### 2.5. Statistical Analysis

All measurements were repeated three times, and the results are expressed as the mean ± standard deviation. One-way analysis of variance (ANOVA) followed by Duncan’s multiple range test using SPSS 23.0 (SPSS Inc., Chicago, IL, USA) was employed to identify significant differences between the samples. The statistical significance was set at *p* < 0.05.

## 3. Results and Discussion

### 3.1. Microbial Levels

As shown in Figure 1, the inoculation ratio had a significant effect on the growth of the microorganisms. When the inoculation ratio of LAB to kombucha was 1:1, the concentration of lactic acid bacteria (LAB), acetic acid bacteria (AAB), and yeast exceeded 6.60 log CFU/mL (7.45, 6.65, and 6.81 log CFU/mL, respectively), thus producing the highest product quality. This is because kombucha is a symbiotic system composed of three types of microorganisms (LAB, AAB, and yeast). They are interdependent, mutually restricted, and symbiotic during the fermentation process, allowing them to hydrolyze the sucrose in the culture medium to form fructose and glucose, followed by further fermentation to produce CO_2_ and alcohol. The AAB in kombucha are weak oxidizers that can oxidize alcohol to produce low levels of acetic acid and oxidize glucose to produce gluconic acid [18].

In this symbiotic system, when the yeast species and AAB multiply, the cells metabolize the surrounding nutrients, creating favorable conditions for the growth of LAB [19]. However, the gradual accumulation of lactic acid during the fermentation of kombucha inhibits the growth and survival of yeast. In addition, when the alcohol reaches a certain volume fraction, the growth of LAB is also inhibited. Therefore, the inoculation ratio directly affects the quality of the finished product by determining whether the collaborative fermentation by the symbiotic organisms can occur [20].

### 3.2. Microbial Sequencing, Splicing, and Diversity Analysis

The biological classification of the microbial communities in the soy milk samples was conducted using the sequencing and filtering of their genetic data (Table 1). According to the analysis of the samples, there were between 78,093 and 114,893 clean reads after the quality control filtering of the original reads. After the overlapping reads were spliced together, gene prediction was conducted using Prodigal, and a non-redundant gene set was obtained. The GC content ranged from 49% to 60%, which can be used for the bioinformatics analysis [21].

Four alpha-diversity indices were employed to assess the diversity of the microorganisms in the soy milk samples: the Shannon, Simpson, and Chao1 indices and coverage. For the Shannon and Simpson indices, higher values reflect higher species diversity, while a higher Chao1 index indicates higher species richness. Coverage refers to the coverage of low-abundance OTUs in each sample, with higher values indicating a higher probability that a sequence in the sample will be observed [22]. As shown in Table 2, the bacterial Shannon and Simpson indices for samples K and LK1.0 were relatively high, indicating a high bacterial OTU diversity. Samples LK0.5, LK1.0, LK2.0, and K also had a high bacterial Chao1 index, illustrating that they contained a higher number of bacterial OTUs, while sample L had the lowest Chao1 index. For the fungal community, the Shannon and Simpson indices for samples K, LK0.5, and LK1.0 were high, while the difference in the Chao1 index among the samples was not significant, which suggested that these samples had similar types of fungal OTUs [23]. The coverage values in Table 2 are all 1.00, indicating that the sequencing generally detected the low-abundance OTUs and that the sequencing results were representative of the actual bacterial and fungal communities in the samples.

### 3.3. Common and Unique Operational Taxonomic Unit Analysis

In this study, the number of OTUs with a 97% sequence identity was counted as another measure of the species richness of the samples. The OTUs found across all samples and those found only in a single sample were displayed using a Venn diagram [21]. As shown in Figure 2a, kombucha alone had the richest variety of bacterial OTUs, with 91 species detected. Among the samples containing both kombucha and LAB, the number of bacterial species was highest when the ratio was 1:1 (i.e., in sample LK1.0), with 82 species in total. Overall, only 10 bacterial species were found in all samples. In the fungal community, the soy milk sample containing kombucha only was also the most species-rich (Figure 2b), with 71 species detected. In the samples containing both kombucha and LAB, the species richness was highest for LK0.5 (67 species). On the whole, 23 fungal species were found in all of the samples.

These results indicate that joint fermentation by kombucha and LAB in soy milk reduces the number of microbial species typically found in kombucha. On the one hand, this may be because the soybean milk matrix itself would be unsuitable for the growth of some microorganisms due to its carbon sources and nutritional characteristics. On the other hand, the addition of LAB has an antagonistic effect on some bacterial groups. In order to determine which species were susceptible to these mechanisms, the species abundance of each sample was analyzed.

### 3.4. Abundance and Dominant Strain Analysis

The present study employed QIIME software to classify the OTUs and produce species profiling maps and histograms for each sample at the phylum and genus levels. The samples contained a total of eight phyla: *Ascomycota*, *Basidiomycota*, *Mucoromycota*, *Firmicutes*, *Actinobacteriota*, *Proteobacteria*, *Bacteroidetes*, and *Fusobacteria*. In sample LK1.0, *Ascomycota* (99.94%), *Proteobacteria* (53.52%), and *Firmicutes* (41.72%) were the most dominant in terms of relative abundance (Figure 3a,b). At the genus level, the preponderant genera were *Lactobacillus* (41.58%), *Acetobacter* (42.39%), *Zygosaccharomyces* (38.89%), and *Saccharomyces* (35.86%) in sample LK1.0 (Figure 3c,d).

These results are similar to those reported by Barbosa et al. [6], who used gene sequencing to analyze a kombucha fermentation broth and found that *Komagataeibacter* and *Zygosaccharomyces* were the most common bacterial and fungal genera. However, in the present study, significantly fewer species were found at the genus level in the samples in which bean milk served as a fermentation substrate when compared with the numbers reported for a kombucha fermentation broth in previous research [8], suggesting that the bean milk matrix exhibits strain selectivity. The metabolic pathways of microorganisms can generally be classified into carbohydrate, protein, and lipid metabolism. The results of the present study indicate that, due to the limited range of carbon sources, proteins, and fat available, not all of the microorganisms in the kombucha could thrive in the soy milk and that the bean milk matrix selects for microorganisms that are the most suitable for its fermentation [24].

Multistage species composition maps were created to illustrate the composition of the microbial community at the realm, phyla, class, order, family, genus, and species levels in the samples. The results showed that the dominant species were *Acetobacter gluconicum*, *Lacticaseibacillus rhamnosus, L. acidophilus*, *Zygosaccharomyces bailii,* and *Saccharomyces cerevisiae* (Figure 4a–e), which represent a combination of LAB, yeast, and AAB. In the symbiotic kombucha community, yeast can hydrolyze sucrose to form glucose and fructose and produce ethanol during fermentation. Ethanol and glucose are then utilized by AAB and converted into acetic acid and gluconic acid, while LAB utilize yeast metabolites [18,19]. FOS was employed as an additional carbon source for microbial fermentation in the present study and had a stimulating effect on microbial growth. This is probably due to the higher levels of fructose released from its partial hydrolysis, which was then metabolized as an additional carbon and energy source [25]. In yeast and LAB co-fermentation systems, the yeast promotes the growth of LAB by producing gluconate, fructose, amino acids, fatty acids, or the precursors required for their synthesis [26].

### 3.5. Cluster Analysis

Cluster analysis was conducted using QIIME (v1.9.1) software. An iterative algorithm was used to sample and analyze 75% of the sequences from each sample based on weighted and unweighted species classification and abundance information. The final cluster tree was obtained after 100 iterations [27].

As shown in Figure 5a,b, the difference between samples L and K was significant for bacteria. In addition, of the co-fermentation samples, LK1.0 and LK2.0 had a high similarity, with little difference in the composition and species abundance of their microbial communities. These results are consistent with those of the abundance analysis (Section 3.4). The results for the fungal communities are similar to those for bacteria, with a high similarity between LK1.0 and LK2.0. These results indicate that the amount of LAB used for fermentation affects the growth and metabolism of the kombucha microflora to a certain extent, and the effect of higher LAB levels was more pronounced in the range of higher levels tested in this study (i.e., a LAB/kombucha ratio of 0.5–2.0). This could be because the lactic acid and other organic acids produced by LAB not only affect the flavor of food but also inhibit the growth of putrefactive and pathogenic bacteria due to the acidification of the fermentation environment. Some *Lactobacillus* are able to produce broad-spectrum bacteriocins, thus inhibiting the growth of some Gram-positive and Gram-negative bacteria and fungi and maintaining the balance of the microbial system [28].

### 3.6. Electronic Nose and Tongue Analysis

The complex data generated by electronic nose and electronic tongue technology combined with multivariate statistical analysis are helpful for the digital evaluation of taste and odor indicators for dairy products, allowing for the quick and efficient identification, classification, and appraisal of samples [29].

The detection results of the electronic nose are presented in Figure 6a. For the fermented soy milk containing kombucha (sample LK1.0), the response to W1C, W5S, W5C, and W2S was significantly enhanced, but that to W3C and W6S did not change significantly. As shown in Figure 6b, there was a positive correlation among five responses (W3C = ammonia, sensitive to aromatic components; W2S = sensitive to alcohols, aldehydes, and ketones; W5C = aromatic components of short-chain alkanes; W2W = aromatic composition, sensitive to organic sulfide; and W3S = sensitive to long-chain alkanes). The samples were widely distributed, indicating that they could be distinguished by the variables. The electronic tongue detection results are displayed in Figure 6c. For the fermented soy milk samples containing kombucha, the response to sourness was significantly higher, while the response to bitterness and aftertaste-B decreased markedly. Figure 6d showed that astringency, umami, and aftertaste-A were positively correlated, and the samples again had a wide distribution.

As shown in Figure 6e, microbial metabolism exhibited a strong correlation with the components identified by the electronic tongue and nose (W1C, W5S, W5C, W2S, and sourness). Previous studies have shown that protease, peptidase, and peptide transporters related to protein hydrolysis are typically associated with the presence of specific plasmids in a strain. The strong aromatic substances produced by acid-producing microbes may be acids, ketones, and amines produced via protein hydrolysis during the rapid metabolism of sugars [30]. The carbohydrate metabolic pathways for multi-strain fermentation systems generally include the TCA cycle, pyruvate metabolism, pentose phosphate, and glycolysis. The intermediate metabolites of these pathways provide the raw materials for the synthesis of many substances, including aromatic amino acids. At the same time, the TCA cycle and pyruvate metabolism pathway can also produce high levels of citric acid, acetic acid, and butyric acid [26].

The increase in the sourness of the samples was likely to be related to microbial growth and metabolism, particularly the metabolism of carbohydrates, proteins, and other substances. This also indicates that, during the fermentation process, the metabolites formed by the symbiotic microorganisms in kombucha may undergo many subsequent biochemical reactions, including the formation of alcohols and aldehydes and the conversion of these to acids and esters [7]. Indeed, the secondary metabolites produced by microorganisms have been identified as the main flavor components of kombucha in previous research [31].

### 3.7. Volatile Flavor Component Analysis

A major factor influencing the commercial prospects of soy milk is its unique taste. The beany flavor of soy milk had been attributed to 12 major volatile compounds, including hexanal, benzaldehyde, acetic acid, and nonanal [32]. As shown in Figure 7a, a total of 40 volatile flavor compounds, including aldehydes, alcohols, esters, ketones, acids, and other compounds, were detected across all of the fermented soy milk samples. *Lactobacillus* and other microorganisms are known to produce various alcohols during amino acid metabolism. Compared with sample S (unfermented soybean milk), hexanal and 1-octene-3-ol in sample LK1.0 were significantly decreased, but not completely eliminated. The content of hexanol in the fermentation system of kombucha and LAB decreased from 30.16% to 8.74%. Hexanal and other volatile flavor substances produced during fermentation with kombucha can bind to proteins through hydrophobic bonds, with longer carbon chains in the protein, leading to a higher binding constant, which may be the reason why some of the soybean flavor remained in the present study [33].

Figure 7b showed that, in the fermented soy milk produced with kombucha, carbohydrates, lipids, and proteins were hydrolyzed to produce primary metabolites such as monosaccharides, free fatty acids, and free amino acids under the synergetic action of microorganisms and endogenous enzymes. Microorganisms convert monosaccharides into pyruvic acid through glycolysis and then into flavor substances, such as short-chain organic acids, alcohols, and carbonyl compounds. Free fatty acids can also react with alcohols in the presence of esterase to produce esters, while unsaturated fatty acids are prone to oxidation, generally producing ketones, aldehydes, and alkanes. Aldehydes can be reduced and oxidized to generate alcohols and acids and then esterified to generate esters [3,7].

Aromatic amino acids, branched amino acids, and methionine produced during the fermentation process are converted by *Zygosaccharomyces* and *Saccharomyces* into higher alcohols via the Ehrlich pathway under the action of transaminase, decarboxylase, and dehydrogenase, while acetate esters are synthesized from higher alcohols or ethanol and acetyl-coenzyme A [7]. In addition to this, Daenen et al. [34] screened *Saccharomyces* strains from the kombucha fungus and found that the relatively high hydrolase activity of *Saccharomyces cerevisiae* and *Brettanomyces custersii* led to the greater release of specific volatiles from hop glycosides during fermentation, such as linalool, methyl salicylate, and 1-octene-3-ol.

Acetic acid, citric acid, and other organic acids produced by *Acetobacter and*
*Gluconacetobacter* during soy milk fermentation gave the milk a sour taste. In addition, the α-acetyl lactic acid generated from citric acid can be transformed into diacetyl during the fermentation process, giving fermented soy milk a unique flavor [35]. LAB form an important group of bacteria that produce organic acids and are associated with changes in volatile aromatic substances. In particular, *Lactococcus* and *Acidobacteria* can promote the production of volatile acids, alcohols, and esters [36], while *Lactobacillus* can promote the synthesis of esters from organic acids and alcohols, producing fermented soy milk with fruity and floral aromas [37]. Huang et al. [3] found that several new flavor compounds were generated during co-fermentation with yeast and LAB, including ethyl acetate and 2-phenylethanol. Thus, the symbiotic interaction between kombucha and LAB can generate many new aromatic substances that have not appeared in soy milk.

## 4. Conclusions

In this study, soybean was used as a raw material for fermentation using kombucha. HTS and GC-MS were used to study the microbial community composition and flavor formation in fermented soy milk samples produced with different fermentation times and various proportions of the fermentation starters. The levels of yeast, LAB, and AAB were highest in the soy milk produced after 42 h of fermentation with a mass ratio of LAB to kombucha of 1:1. The soy milk substrate exhibited good strain selectivity. In the soy milk samples containing kombucha (LK1.0), the dominant genera were *Lactobacillus*, *Acetobacter*, *Zygosaccharomyces,* and *Saccharomyces*. In addition to improving the sourness, fermentation reduced the levels of unpalatable flavor substances in the soy milk and led to the formation of unique aroma compounds, such as 2,5-dimethylbenzaldehyde and linalool. The use of a variety of lactic acid bacteria can improve the flavor richness of kombucha-fermented soy milk. This study thus demonstrates the feasibility of using multi-strain co-fermentation to improve the flavor of fermented soybean products and provides a theoretical basis for the flavor formation mechanisms associated with fermented food and a reference for flavor simulation and innovation in the field of plant-based foods in the future. In future research, in order to clarify the internal relationship between bacteria metabolism and flavor in more detail, the differences in and reasons for dynamic changes in flavor substances in the process of multi-bacterial co-fermentation are still worthy of systematic research and analysis.

## Figures and Tables

**Figure 1 foods-12-00884-f001:**
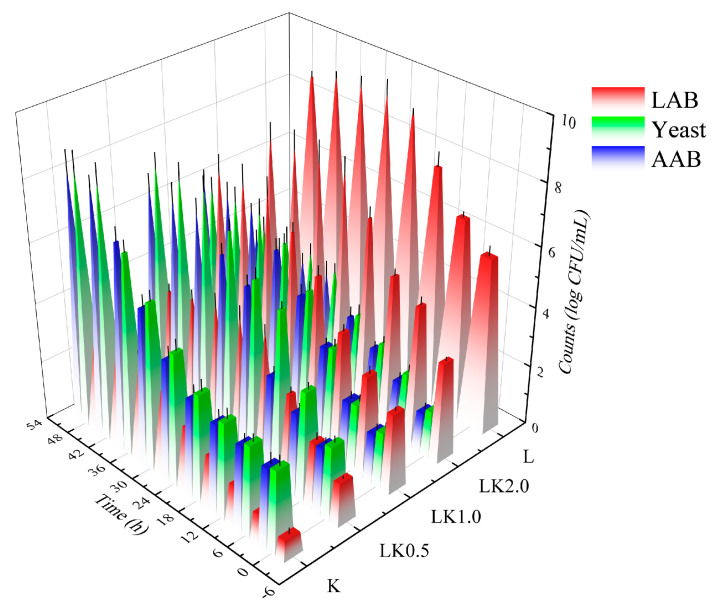
Changes in the microbial content of fermented soy milk samples. K = kombucha fermentation; LK0.5, LK1.0, and LK2.0 = Lactobacillus bacteria (LAB) fermentation combined with kombucha at mass ratios of 0.5, 1.0, and 2.0, respectively; L = LAB fermentation.

**Figure 2 foods-12-00884-f002:**
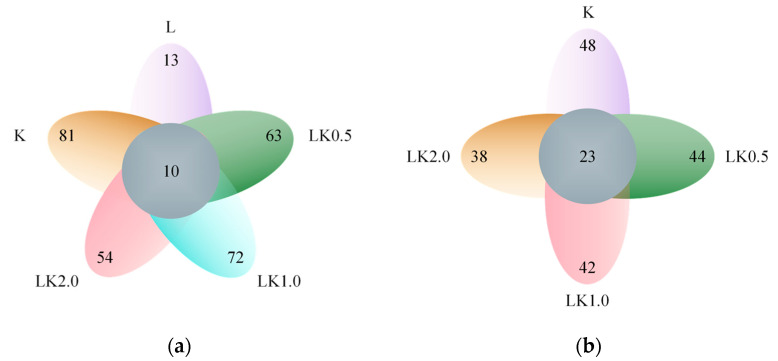
Venn diagrams for (**a**) bacteria and (**b**) fungi showing the number of species in each sample and those shared by all samples.

**Figure 3 foods-12-00884-f003:**
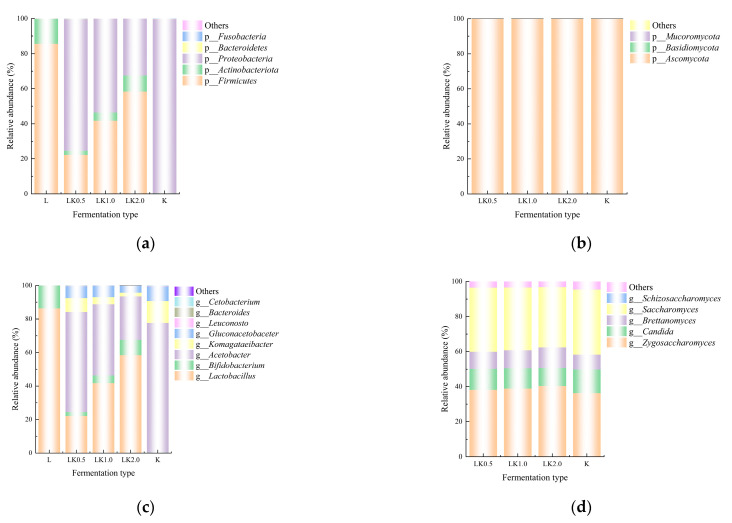
Relative abundance at the (**a**) bacterial phylum, (**b**) fungal phylum, (**c**) bacterial genus, and (**d**) fungal genus levels.

**Figure 4 foods-12-00884-f004:**
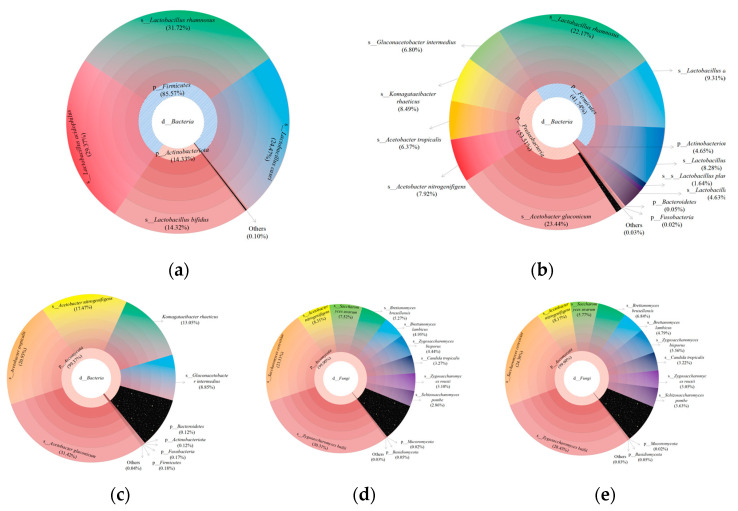
Multistage bacterial species composition for the soy milk samples (**a**) L, (**b**) LK1.0, and (**c**) K. Multistage fungal species composition for the soy milk samples (**d**) LK1.0 and (**e**) K.

**Figure 5 foods-12-00884-f005:**
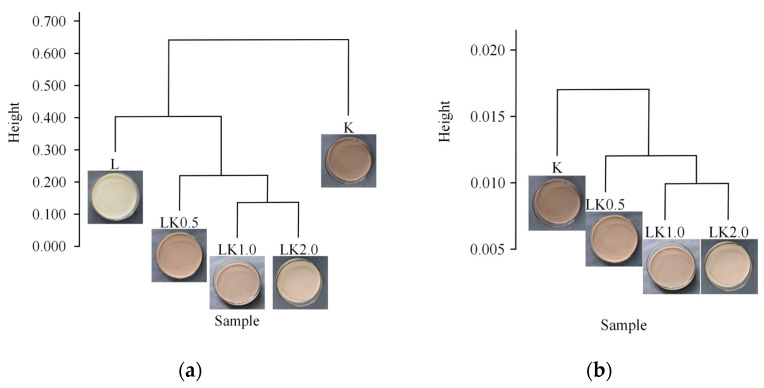
Clustering trees for (**a**) bacteria and (**b**) fungi.

**Figure 6 foods-12-00884-f006:**
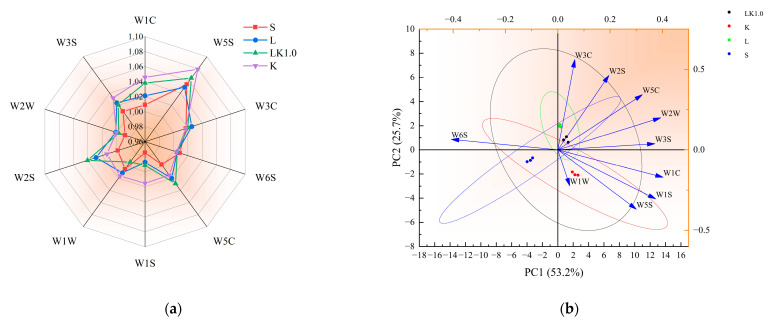
(**a**) Electronic nose radar diagram. (**b**) Principal components for the electronic nose data. (**c**) Electronic tongue radar diagram. (**d**) Principal components for the electronic tongue. (**e**) Correlations among the microbes, smell, and taste.

**Figure 7 foods-12-00884-f007:**
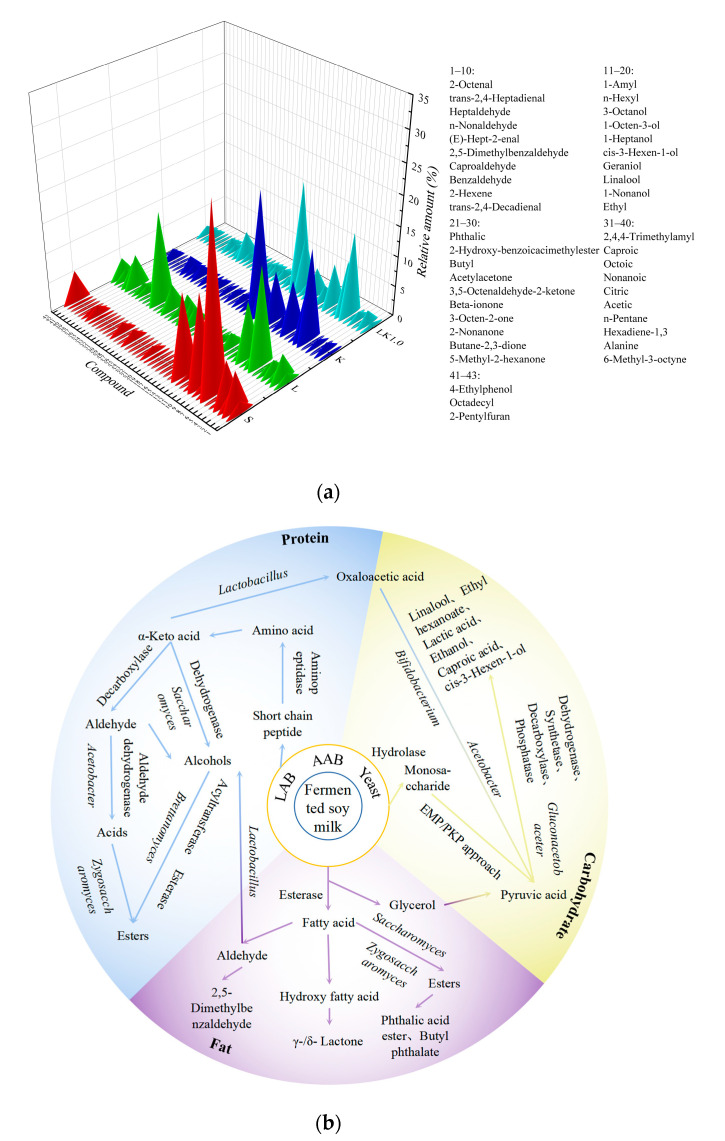
(**a**) Volatile flavor components in fermented soy milk. S = unfermented soy milk. (**b**) Main formation pathways for the flavor substances, blue: Protein metabolism; yellow: Carbohydrate metabolism; purple: Fat metabolism.

**Table 1 foods-12-00884-t001:** Sample sequencing results.

Sample	Bacteria				Fungi			
	Clean_Paired_Reads	GC (%)	Num_Len	Avg_Len	Clean_Paired_Reads	GC (%)	Num_Len	Avg_Len
L	78,093	52	64,595	430	ND	ND	ND	ND
LK0.5	97,688	49	81,353	429	86,035	59	73,704	281
LK1.0	101,065	50	84,554	428	98,194	60	89,125	284
LK2.0	102,385	53	82,093	423	94,918	60	85,786	283
K	105,886	50	86,528	425	114,893	60	104,350	284

ND: “not detected” within the detection range.

**Table 2 foods-12-00884-t002:** Bacterial and fungal diversity indices.

Sample	Bacteria				Fungi			
	Shannon	Simpson	Chao1	Coverage	Shannon	Simpson	Chao1	Coverage
L	0.61	0.19	86.25	1.00	ND	ND	ND	ND
LK0.5	0.94	0.39	113.80	1.00	0.63	0.16	54.72	1.00
LK1.0	1.09	0.43	110.62	1.00	0.58	0.19	51.46	1.00
LK2.0	0.93	0.34	108.94	1.00	0.37	0.14	48.93	1.00
K	1.46	0.58	102.11	1.00	0.74	0.22	56.35	1.00

ND: “not detected” within the detection range.

## Data Availability

Data is contained within the article.

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
