# Peer review of "Microbial Diversity and Flavor Regularity of Soy Milk Fermented Using Kombucha"

_foods, 2023, doi:10.3390/foods12040884_

Round 1

Reviewer 1 Report

COMMENTS TO THE AUTHORS:

This is an interesting manuscript describing soy milk fermentation using kombucha. But the following points need to be done by the authors:

1.     Abstract: Please support the results with some quantitative data.

2.     Line 93: Are these strains lab strains or isolated with a specific origin (e.g. food sample)? As lab strains may behave differently from isolated ones, this would be something that has to be considered.

3.     Line 99: Mention why FOS and sucrose were added to the soymilk and autoclaving the mixture didn’t have a negative effect on the carbohydrates?

4.     Line 106: Were the lactobacillus strains inoculated together? Why not separately? Please comment.

5.     Line 107: Mention the g force of the centrifuge.

6.     Line 278: Lactobacillus rhamnosus, I now known as Lacticaseibacillus rhamnosus. A tool can be used to quickly search for new names of bacteria: (http://lactobacillus.ualberta.ca/).

7.     Paragraphs are too long. Check paragraphs extension in the manuscript.

8.     Conclusion: what are the future of your findings? Conclusion is not insightful, what are suggestions?

9.     English language of paper needs to be revised (English Editing Service) therefore I don't suggest word by word editions.

Author Response

Reviewer #1: The manuscript by Peng et al. is about preparation of soy milk by a combination of fermented technique. The authors used a fermented product, Kombucha, along with lactic acid bacteria to produce another fermented product. The manuscript is interesting and minor corrections are required.

Authors response: We thank the reviewer for their complimentary comments and contribution to improvements of this manuscript. We have carefully revised, supplemented, and explained in accordance with the reviewers' comments and questions, and checked the language of the full text. The responses to the detailed comments have been listed below.

  1. Abstract: Please support the results with some quantitative data.

Authors response: Thank you for your good advice. We have supported the results with quantitative data as follows:

Line 26-33: “In the fermented soy milk produced by kombucha and LAB, the dominant bacterial genera were Lactobacillus (41.58%) and Acetobacter (42.39%), while the dominant fungal genera were Zygosaccharomyces (38.89%) and Saccharomyces (35.86%). After 42 h, the content of hexanol in the fermentation system of Kombucha and LAB decreased from 30.16% to 8.74%, while flavor substances such as 2,5-dimethylbenzaldehyde and linalool were produced”. 

  1. Line 93: Are these strains lab strains or isolated with a specific origin (e.g. food sample)? As lab strains may behave differently from isolated ones, this would be something that has to be considered.

Authors response: Thank you for your good advice. The LAB used in this study were customized by our long-term school-enterprise cooperation enterprises and developed for the fixed projects of school-enterprise. The strains are cultivated professionally by the manufacturer, and have been specially tested in the aspects of strain vitality, stability and fermentation characteristics. In addition, in our previous research and published papers, the fermentation effect of these strains has also been verified. After fermentation, the physical and chemical properties of soybean milk, the content of aglycone isoflavones and vitamin have been improved. Thank you again for your suggestion. We have supplemented the source of strains as follows:

Line 119-121: “Lactic acid bacteria (LAB), including Lactobacillus acidophilus, L. bifidus, L. rhamnosus, and L. casei (commercially purchased, 1:1:1:1), were cultured at Xi'an Dongfeng Biotechnology Co., Ltd”.

  1. 3. Line 99: Mention why FOS and sucrose were added to the soymilk and autoclaving the mixture didn’t have a negative effect on the carbohydrates?

Authors response: Thank you for your good advice.

(1) Soy milk has low acidification rate and slow growth of probiotic bacteria which take longer time to complete the fermentation and produces undesirable changes in the product which is not acceptable to the consumer. As a common prebiotic, FOS can effectively improve the vitality of probiotics [1]. In addition, incorporation of FOS also improves the sensory profile, physicochemical and rheological characteristics of probiotic fermented products [2].

(2) Sucrose can increase the sweetness, adjust the ratio of sugar and acid, and not make the fermented soy milk taste too sour. besides, sucrose can supplement the lack of carbon source during probiotics fermentation. Therefore, we added FOS and sucrose.

(3) Generally, Sucrose and FOS are still stable at 120 ℃. The sterilization condition of this experiment is to sterilize at 115 ℃ for 15 min. Autoclaving has a slight negative impact on carbohydrates, but has little impact on the theme of this experiment. Autoclaving can also kill heat-resistant spores. This method is also one of the commonly used sterilization methods for soybean milk production [3,4]. Thank you again for your question.

Overall, because FOS and sucrose were added to all samples in the experiment, there would be no variable impact on the experiment. Once again, thank you very much for your comments and suggestions.

  • Fonteles,T. V., Rodrigues, (2018). Prebiotic in fruit juice: processing challenges, advances, and perspectives. Current Opinion in Food Science, 22, 55-61.
  • Hussien,H., Abd, R. H., et al. (2022). The impact of incorporating Lactobacillus acidophilus bacteriocin with inulin and FOS on yogurt quality. Scientific Reports, 12, 13401-13401.
  • Hsin, Y., Kuo, S. H., et al. (2013). Preparation and Physicochemical Properties of Whole-Bean Soymilk. Journal of Agricultural and Food Chemistry, 62, 742-749.
  • Escobedo,A., Loarca, P. G., et al. (2020). Autoclaving and extrusion improve the functional properties and chemical composition of black bean carbohydrate extracts.. Journal of food science, 85, 2783-2791.
  1. 4. Line 106: Were the lactobacillus strains inoculated together? Why not separately? Please comment.

Authors response: Thank you for your good advice. We must apologize for our ambiguous expression. In fact, we inoculated the strains separately during the experiment. Thank you very much for pointing out our problem in time. We have revised this sentence as follows:

Line 128-132: “Different secondary activation solutions (4% v/v) were inoculated into soy milk samples in turn and cultured at 32 ℃ in an incubator to obtain fermented soy milk with different fermentation times. The inoculation sequence of the strains was Lactobacillus acidophilus, L. bifidus, L. rhamnosus, L. casei and kombucha strains”.

  1. 5. Line 107: Mention the g force of the centrifuge.

Authors response: Thank you for your good advice. Sorry, we didn't find the relevant question on line 107. Is it on line 117? We have revised this sentence on line 117 as follows. In addition, in order to avoid the same problem, we also carefully checked and revised the full text.

Line 154-155: “Aliquots (1.5 mL) of the fermented soy milk samples were centrifuged (1500 g, 4 ℃, 10 min) and 567 μL of TE buffer was added”. 

  1. 6. Line 278: Lactobacillus rhamnosus, I now known as Lacticaseibacillus rhamnosus. A tool can be used to quickly search for new names of bacteria: (http://lactobacillus.ualberta.ca/).

Authors response: Thank you for your good advice. We have revised this sentence as follows. In addition, in order to avoid the same problem, we also carefully checked and revised the full text. And we have collected this website, which will provide great help for our next experiment and writing materials. Thank you very much.

Line 324-326: “The results showed that the dominant species were Acetobacter gluconicum, Lacticaseibacillus rhamnosus, L. acidophilus, Zygosaccharomyces bailii, and Saccharomyces cerevisiae”. 

  1. 7. Paragraphs are too long. Check paragraphs extension in the manuscript.

Authors response: Thank you for your good advice. We have checked paragraphs extension and segmented it according to the article structure, such as lines 90, 302, etc, and all corrections and changes were marked up using the “Track Changes” function.

  1. 8. Conclusion: what are the future of your findings? Conclusion is not insightful, what are suggestions?

Authors response: Thank you for your good advice. We have supplemented the future of our findings and suggestions as follows:

Line 489-497: “The use of a variety of lactic acid bacteria can improve the richness of the flavor of Kombucha fermented soy milk. This study thus demonstrated the feasibility of using multi-strain co-fermentation to improve the flavor of fermented soybean products, provided a theoretical basis for the flavor formation mechanisms associated with fermented food and a reference value for flavor simulation and innovation in the field of plant-based foods in the future. In the future research, in order to clarify the internal relationship between bacteria metabolism and flavor in more detail, the differences and reasons of dynamic changes of flavor substances in the process of multi-bacteria co-fermentation are still worth systematic research and analysis”. 

  1. 9. English language of paper needs to be revised (English Editing Service) therefore I don't suggest word by word editions.

Authors response: We would like to thank the reviewers. We apologize for the poor language of our manuscript. We worked on the manuscript for a long time and the repeated addition and removal of sentences and sections obviously led to poor readability. We have accepted the use of a language editing service to proofread the manuscript, and have also involved native English speakers for language corrections. The touch-up certificate was attached. Revised portions are marked in blue and red in the paper. The touch-up certificate is as follows:

Reviewer 2 Report

The manuscript by Peng et al. is about preparation of soy milk by a combination of fermented technique. The authors used a fermented product, Kombucha, along with lactic acid bacteria to produce another fermented product. The manuscript is interesting and minor corrections are required. 

1.     The first line of abstract is too general. Write rationale of this study in 2-3 sentences. Such as, ‘Plant milk is considered as healthy and environmentally sustainable option yet its production …’

2.     L20-22: Revise

3.     In abstract, write about the source of LAB and mention the nature of Kombucha

4.     L43-46: Avoid writing incomplete sentences

5.     L47: Write ‘..Bioinformatic tools..’

6.     L48: Write ‘..next generation sequencing..’

7.     L69: Write full form of HTS

8.     In introduction, write about current method of preparing soy milk. State what advantage the current strategy will provide

9.     L92: Only names are italicized

10.  The composition of LAB, in terms of proportion of various strains/species should be clearly mentioned

11.  Section 2.3.1, more details are needed to grasp the basic procedure

12.  3.3: Revise heading of this section. Write full form of OUT. Do not mention about the figure/diagram

 Author Response

Reviewer #2: This is an interesting manuscript describing soy milk fermentation using kombucha. But the following points need to be done by the authors.

Authors response: We thank the reviewer for their complimentary comments and contribution to improvements of this manuscript. We have carefully revised, supplemented, and explained in accordance with the reviewers' comments and questions, and checked the language of the full text. The responses to the detailed comments have been listed below.

  1. The first line of abstract is too general. Write rationale of this study in 2-3 sentences. Such as, ‘Plant milk is considered as healthy and environmentally sustainable option yet its production …’

Authors response: Thank you for your good advice. We have rephrased this sentence as follows:

Line 13-18: “Plant milk is considered as healthy and environmentally sustainable option. However, due to the low protein content of most plant milk and the difficulty of flavor acceptance by consumers, its production scale is usually limited. Soy milk is a kind of food with comprehensive nutrition and high protein content. In addition, kombucha is naturally fermented by acetic acid bacteria (AAB), yeast, lactic acid bacteria (LAB) and other microorganisms, the microorganisms in its system can improve the flavor characteristics of food”.

  1. L20-22: Revise

Authors response: Thank you for your good advice. We have revised this sentence as follows:

Line 26-28: “In the fermented soy milk produced by kombucha and LAB, the dominant bacterial genera were Lactobacillus (41.58%) and Acetobacter (42.39%), while the dominant fungal genera were Zygosac-charomyces (38.89%) and Saccharomyces (35.86%)”.

  1. In abstract, write about the source of LAB and mention the nature of Kombucha

Authors response: Thank you for your good advice.

(1) The LAB used in this study were customized by our long-term school-enterprise cooperation enterprises. The strains are cultivated professionally by the manufacturer, and have been specially tested in the aspects of strain vitality and fermentation characteristics. In addition, in our previous research and published papers, the fermentation effect of these strains has also been verified. After fermentation, the physical and chemical properties of soybean milk, the content of aglycone isoflavones and vitamin have been improved. Due to the limitation of the number of words in the abstract, we added the source of bacteria as "commercially purchased". We have supplemented the source of LAB as follows:

Line 19-20: “In the present study, LAB (commercially purchased) and kombucha were used as fermenting agents for soybean as a raw material to produce soy milk”. 

(2) We have supplemented the nature of Kombucha as follows:

Line 16-18: “kombucha is naturally fermented by acetic acid bacteria (AAB), yeast, lactic acid bacteria (LAB) and other microorganisms, the microorganisms in its system can improve the flavor characteristics of food”. 

  1. L43-46: Avoid writing incomplete sentences

Authors response: Thank you for your good advice. We have revised this sentence as follows:

Line 57-62: “In addition, hexaldehyde and glutaraldehyde (the main source of the beany flavor of soy milk) were produced via the peroxidation of unsaturated fatty acids catalyzed by lipoxygenase, and fermentation can degrade some aldehydes. For example, Huang et al. [3] reported that co-fermentation (lactic acid bacteria combined with Kluyveromyces marxianus) can effectively degrade the aldehydes in goat milk and produce organic acid compounds”.  

  1. L47: Write ‘..Bioinformatic tools..’

Authors response: Thank you for your good advice. We have revised this sentence as follows:

Line 63-64: “With the development of bioinformatic tools, new techniques for food flavor analysis and microbial characterization are emerging”. 

  1. L48: Write ‘..next generation sequencing..’

Authors response: Thank you for your good advice. We have revised this sentence as follows:

Line 64-69: “Indeed, next generation sequencing (macrogenomics, flavoromics, etc.) and flavor analysis (electronic tongue, gas chromatography–mass spectrometry [GS-MS], etc.) technologies have played an important role in exploring the mechanisms associated with the formation of characteristic flavor substances and in clarifying the contribution of microorganisms to this process”. 

  1. L69: Write full form of HTS

Authors response: Thank you for your good advice. We have revised this sentence as follows:

Line 85-89: “Marsh et al. [8] employed high-throughput sequencing technology to investigate the composition of the microbial community in kombucha and found that the two most dominant bacterial genera were Acetobacter and Gluconacetobacter, accounting for up to 90% of the total bacterial population”. 

  1. In introduction, write about current method of preparing soy milk. State what advantage the current strategy will provide

Authors response: Thank you for your good advice.

(1) We have supplemented the current method of preparing soy milk as follows:

Line 50-52: “At present, the preparation method of fermented soy milk mainly include selecting soybeans, grinding soy milk, adding ingredients, homogenizing, sterilizing, inoculating, fermenting, and ripening after refrigeration”. 

(2) Compared with single strain fermentation, this study covers a wider range of microorganisms, including yeast, lactic acid bacteria, acetic acid bacteria and other common strains. Besides, according to our previous experiments and published papers, the kombucha-fermentation process could significantly change the viscoelastic rheological properties of soy milk. Most of the isoflavone glycosides were hydrolyzed to aglycones by β-glucosidase produced during fermentation. At the same time, the kombucha-fermentation process also promoted the formation of unique flavor and bring satisfaction to consumers in taste and flavor. Therefore, this study can more accurately clarify the relationship between the dominant strains and flavor substances in soy milk. We have rewritten the introduction section as follows in order to make these basic considerations and resulting advantages of our strategy clearer.

Line 104-111: “Compared with single strain fermentation, this study covers a wider range of microorganisms, including yeast, lactic acid bacteria, acetic acid bacteria and other common strains. In addition, the fermentation of soy milk with kombucha can not only improve the rheological properties of soymilk, increase the content of vitamins and aglycone isoflavones in soy milk, but also promote the formation of unique flavor and bring satisfaction to consumers in taste and flavor. Therefore, this study can more accurately clarify the relationship between the dominant strains and flavor substances in soy milk”.  

  1. L92: Only names are italicized

Authors response: Thank you for your good advice. The phrase "Lactic acid bacteria (LAB)" has been modified to a non-italic format at Line 119. In addition, in order to avoid the same problem, we also carefully checked and revised the full text.

  1. The composition of LAB, in terms of proportion of various strains/species should be clearly mentioned

Authors response: Thank you for your good advice. We have revised this sentence as follows:

Line 119-121: “Lactic acid bacteria (LAB), including Lactobacillus acidophilus, L. bifidus, L. rhamnosus, and L. casei (commercially purchased, 1:1:1:1), were cultured at Xi'an Dongfeng Bio-technology Co., Ltd”. 

  1. Section 2.3.1, more details are needed to grasp the basic procedure

Authors response: Thank you for your good advice. We have supplemented more details to reflect the basic procedure, including standard bacteria and fungi, qPCR procedure and modified Gompertz equation as follows:

Line 142-152: “The standard curve was generated using the ITS rRNA gene of Pichia pastoris, the 16S rRNA gene of Komagataeibacter xylinus and Lactobacillus sakei. A StepOnePlus Real-time PCR System was used to conduct absolute qPCR analysis of the yeasts, AAB, and LAB in the samples following the method reported by Xiao et al. [12]. RNA was reversely transcribed into CDNA, CDNA was added to the Real time dye PCR PreMIXA2010A0112, and then samples were added to the qPCR Array. The microbial growth curves were calculated by the following modified Gompertz equation.

       (1)

Where N(T), cell number at T; Nmax, maximum cell number; N0, initial cell number; μmax, specific growth rate; L, lag phase; T, sampling time”. 

  1. 3.3: Revise heading of this section. Write full form of OUT. Do not mention about the figure/diagram

Authors response: Thank you for your good advice. We have revised this heading as follows:

Line 269: “3.3. Common and unique Operational Taxonomic Units analysis”. 

Round 2

Reviewer 1 Report

The authors have generated the suggested changes with clear and concrete responses to each of the comments I made. As a result, the manuscript is much improved. I have no objection to publication in Foods.